# Decomposition-based Neural Multi-objective Combinatorial Optimization with Graph-Filter based Multi-Head Attention

## Abstract

Recent decomposition-based neural multi-objective combinatorial optimization (MOCO) methods have achieved remarkable success in solving Multi-Objective Combinatorial Optimization Problems (MOCOPs). However, existing methods either overlook the relationships between weights and instances of the subproblems, or cannot sufficiently capture their relationships. This may lead to poor synergy between weights and instances, which impairs the ability of learning the Pareto fronts. To address this limitation, we propose a novel framework that can more effectively characterize the relationships between weights and instances while mitigating the over-smoothing problem. Specifically, we have designed a cross-attention-based weight embedding model, which treats weight vectors and node vectors as two separate sequences to gain deeper insights into their interdependencies. Additionally, we employ a graph-filter-based multi-head attention module that optimizes attention computations to prevent the loss of critical information during backpropagation. We validated the effectiveness and versatility of our method on several classic MOCOP benchmarks. Experimental results show that our method outperforms existing state-of-the-art approaches with significant superiority and strong generalization capabilities.

## 1 Introduction

Solving multi-Objective Combinatorial Optimization Problems (MOCOPs) has attracted increasing attentions from both computer science and operational research domains, because they have wide-ranging practical applications in fields such as manufacturing Ahmadi et al. (2016), logistics Jozefowiez et al. (2008), and scheduling Ghoseiri et al. (2004). In such problems, a decision-maker must simultaneously balance multiple competing criteria, such as cost, makespan, and environmental impact, etc. Unlike Single-Objective Combinatorial Optimization Problems (SOCOPs) that pursue a single optimal solution, MOCOPs aim to identify a set of Pareto optimal solutions. This set represents the various trade-offs between different objectives, making the problem inherently more complex to solve.

In recent years, neural approaches have emerged as the mainstream methodology for MOCOPs. Most them typically decompose a MOCOP into a set of scalarized subproblems, i.e., single-objective problems, defined by different weight vectors, then solve them using Deep Reinforcement Learning (DRL) to approximate the Pareto front. Early studies employed transfer learning or meta-learning techniques to train or fine-tune a separate model for each subproblem Li et al. (2020); Zhang et al. (2022). As an improvement, PMOCO Lin et al. (2022) utilized a weight-conditioned hypernetwork to dynamically adjust model parameters, enabling a single model to handle all subproblems, yet its effectiveness remain limited when dealing with diverse weight vectors. Recent weight-specific neural MOCO methods, such as CNH Fan et al. (2024) and WE-CA Chen et al. (2025), partly alleviates the impact of varying weights on the model's generalization performance by learning weight-specific representations via a unified model that capture weight-instance interaction for the subproblems, making them the state-of-the-art (SOTA) approaches.

However, current weight-specific neural MOCO methods still struggle to effectively fuse the node and weight information and capture the relationship between them. As in WE-CA, a succinct at-

tention model and an enhanced conditional attention model are used to learn weight-specific node embeddings and weight embedding and node embeddings simultaneously, respectively, with the purpose of capturing the weight-instance interaction. Both instantiations achieved superior performance on classic MOCOPs. However, its limitation lies in the direct fusion of weight and node vectors via conditional attention, which may discard the original node features and make the model susceptible to noise injection and may suffer from training instability.

To address this problem, we propose a weight embedding module based cross-attention, which treats the weight and node vectors as separate sequences, enabling it to gain a more profound understanding of their relationship. Nevertheless, directly incorporating cross-attention inevitably increases encoder complexity and aggravates the *over-smoothing* problem, where node features become indistinguishable after multiple propagation layers. To mitigate this issue, we further integrate the Graph-Filter-based Multi-Head Attention (GFMA) module into our framework. This module can effectively mitigate the *over-smoothing* by introducing a meticulously designed Graph-Filter composed of an identity term and two matrix polynomials Choi et al. (2024).

In short, we design **GF-MOCA**, a **G**raph-**F**iltered **M**ulti-**o**bjective **C**ross-**A**ttention Framework. It integrates cross-attention for modeling weight–node interactions and the GFMA module to preserve structural sensitivity during multi-head attention meanwhile mitigating the *over-smoothing* problem, thereby allowing more accurate and robust interaction modeling.

Our main contributions are: (1) We design a novel cross-attention-based weight embedding module to address the flawed information flow and noise introduction inherent in existing weight-based neural MOCO methods. This module treats weight and node vectors as separate sequences to gain a more profound understanding of their relationships, thereby enhancing the overall solution quality. (2) We introduce the Graph-Filter into our framework to overcome the *over-smoothing* issues caused by the increasing complexity of encoders, thereby enabling the efficient integration of sophisticated components like cross-attention. (3) We demonstrate the effectiveness and versatility of our method through extensive experiments on various MOCOPs. The results indicate that proposed method not only outperforms the state-of-the-art neural MOCO methods, but also exhibits generalization capabilities across problems of different scales and types.

## 2 RELATED WORK

### 2.1 TRADITIONAL MOCO METHODS

Traditional MOCO methods can be roughly divided into two categories: exact and heuristic methods. The former can find exact Pareto optimal solutions for MOCOPs. However, for large-scale instances or problems with a large number of Pareto-optimal solutions, the computational cost of exact methods often increases exponentially, quickly rendering them impractical Halffmann et al. (2022); Wu et al. (2022); Bergman et al. (2022). In contrast, heuristic methods can find approximate ones within an acceptable time frame.

Consequently, the research focus has gradually shifted towards heuristic methods Jozefowiez et al. (2008); Zajac & Huber (2021). Among these methods, Multi-Objective Evolutionary Algorithms (MOEAs) are widely used due to their effectiveness Tian et al. (2021); Falcón-Cardona et al. (2021). Existing MOEAs typically fall into two categories: Dominance-based MOEAs, represented by NSGA-II Deng et al. (2022); Seada & Deb (2015); Deb et al. (2002); Deb & Jain (2013), and decomposition-based ones, represented by MOEA/D Zhang & Li (2007); Ke et al. (2014); Qi et al. (2014); Yuan et al. (2015); Fang et al. (2020). Despite the great success of MOEAs, they often require extensive manual engineering, such as selecting crossover, mutation, and selection operators, as well as tuning specific hyperparameters Tian et al. (2021); Xie et al. (2022); Yi et al. (2020), which limits their generalization performance on various instances and problems.

### 2.2 NEURAL MOCO METHODS

In recent years, researchers have begun extending neural methods for solving MOCOPs, and decomposition-based approaches have emerged as the mainstream. These decomposition-based neural MOCO methods generally fall into two paradigms: multi-model and single-model. Multi-model methods train or fine-tune an independent neural network model for each decomposed subproblem

Li et al. (2020); Zhang et al. (2021; 2022); Chen et al. (2023). To accelerate convergence, researchers have adopted techniques such as parameter transfer Wu et al. (2019) or meta-learningChen et al. (2023). However, this paradigm faces challenges of high training overhead and the need to maintain multiple models. In contrast, single-model methods use only a single neural network model to handle all subproblems, such as the DRL frameworks based on hypernetworks Lin et al. (2022); Li et al. (2023); Wu et al. (2024) and conditional neural heuristics that take weight vectors as input Chen et al. (2025); Fan et al. (2024); Lin et al. (2022); Wang et al. (2023). These methods always exhibit outstanding generalization performance across problem sizes and distribution, but are highly dependent on and sensitive to the performance of the used model.

## 3 PRELIMINARY

### 3.1 MULTI-OBJECTIVE COMBINATORIAL OPTIMIZATION PROBLEM

Without loss of generality, an MOCOP can be formally defined as:

$$\min_{\pi \in \mathcal{X}} \mathbf{F}(\pi) = (f_1(\pi), f_2(\pi), \dots, f_\kappa(\pi)) \tag{1}$$

where $\mathbf{F}$ is the objective vector comprising $\kappa$ conflicting objective functions, $\pi$ is a feasible solution in the feasible solution space $\mathcal{X}$. Due to inherent conflicts among objectives, there is typically no single solution that is simultaneously optimal for all objectives. Therefore, the goal is to find a set of solutions, which are so-called Pareto set (PS), that represent different trade-offs among these competing objectives. The Pareto-related properties of solutions are defined as follows.

**Definition 1: Pareto Dominance** For any two solutions $\pi$ and $\pi'$ in the feasible solution space $\mathcal{X}$, we say $\pi$ dominates $\pi'$ (denoted as $\pi \prec \pi'$) if and only if both of the following conditions hold: for all objectives $i \in \{1, \dots, \kappa\}$, $f_i(\pi) \leq f_i(\pi')$; and the objective vector $\mathbf{F}(\pi) \neq \mathbf{F}(\pi')$.

**Definition 2: Pareto Optimality** A solution $\pi^* \in \mathcal{X}$ is called Pareto optimal if it is not dominated by any other solution in the feasible solution space $\mathcal{X}$. Consequently, the Pareto set $\mathcal{P}$ is defined as the set of all Pareto optimal solutions: $\mathcal{P} = \{\pi^* \in \mathcal{X} \mid \nexists \pi \in \mathcal{X} : \pi \prec \pi^*\}$. The Pareto front $\mathcal{F}$ is defined as the image of the Pareto set in the objective space: $\mathcal{F} = \{\mathbf{F}(\pi^*) \mid \pi^* \in \mathcal{P}\}$.

### 3.2 DECOMPOSITION-BASED METHODS

The core idea of the decomposition strategy Zhang & Li (2007) is to decompose a complex MOCO problem into $N$ relatively simple scalarized subproblems, each of which can be treated as a SOCOP. This process relies on a predefined set of weight vectors $\{\lambda_1, \lambda_2, \dots, \lambda_N\}$, where each weight vector $\lambda \in \mathbb{R}^M$ satisfies the conditions $\lambda_m \geq 0$ for all $m$, and $\sum_{m=1}^{M} \lambda_m = 1$.

The most representative decomposition method used in this paper is the Weighted Sum (WS). The scalarized objective function, i.e., $g(x|\lambda)$, of a subproblem characterized by $\lambda$ can be defined as:

$$\min_{\pi \in \mathcal{X}} g_{ws}(\pi|\lambda) = \sum_{m=1}^{M} \lambda_m f_m(\pi) \tag{2}$$

Here, the subscript $ws$ indicates the weighted sum decomposition method.

Within the framework of neural combinatorial optimization, researchers employ neural CO methods, such as POMO Kwon et al. (2020), to solve these subproblems defined by different weight vectors $\lambda$, thereby approximating the complete PS. The solving process is formulated as a Markov Decision Process (MDP). Specifically, a feasible solution $\pi$ is represented as a sequence of decisions of length $T$, i.e., $\pi = \{\pi_1, \dots, \pi_T\}$. For a given problem instance $s$ and a weight vector $\lambda$, the model's objective is to learn a stochastic policy $P(\pi|s, \lambda)$ that can generate high-quality solutions. This policy constructs a solution sequentially as follows:

$$P(\pi|s, \lambda) = \prod_{t=1}^{T} P_\theta(\pi_t|\pi_{1:t-1}, s, \lambda) \tag{3}$$

The reward of DRL reward is typically set to the negative of the scalarized objective value, i.e., $-g(\pi|\lambda, s)$. By maximizing this reward, the policy network $P_\theta$ is trained to generate the corresponding optimal solutions for subproblems characterized by different weight vectors.

## 4 METHOLOGY

GF-MOCA is a learning-based framework designed to solve the MOCOPs. It employs a cross-attention based fusion module to dynamically model the regulatory effect of the weight vector $\lambda$ on node vector representations, thereby learning the relationship between them. Additionally, to mitigate the over-smoothing issues caused by the complexity of encoders, the GFMA model is incorporated into GF-MOCA. In Section 4.1 we introduce the architecture of the GFMA module, which serves as a submodule within the cross-attention model, and Section 4.2 presents the complete model design.

### 4.1 GRAPH-FILTER-BASED MULTI-HEAD ATTENTION

To mitigate the over-smoothing issues, we propose a Graph-Filter-based multi-head attention mechanism, as depicted in Figure 1. This module receives the output from the cross-attention module, denoted as $X^{(l-1)} = [h_1^{(l-1)}; \ldots; h_n^{(l-1)}; h_\lambda^{(l-1)}]$. Then it generates the query ($q$), key ($k$), and value ($v$) vectors through a standard linear layer. Subsequently, a processed attention matrix, $\tilde{H}_{GFMA}$, is produced by the Graph-Filter Module, which will be introduced in the following section. Finally, this GFMA attention matrix $\tilde{H}_{GFMA}$ is used to perform a weighted sum on the value vectors $v$ to produce the output for each attention head.

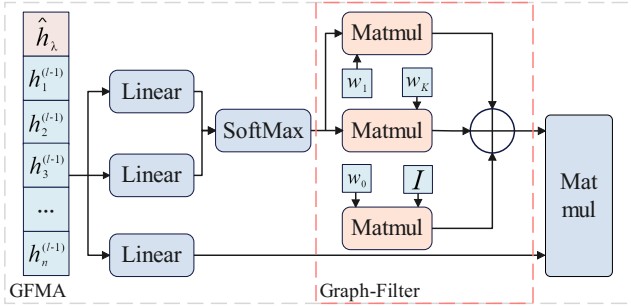

Figure 1: Graph-Filter-based Multi-Head Attention Model.

**Graph-Filter Module.** Given a query $q$, key $k$, and value $v$, the initial attention weight matrix, which we denote as $\overline{A}$, is calculated through dot product and a Softmax operation. Then, the attention weight matrix is passed through a Graph-Filter, as defined in Eq. (4), which utilizes the two lowest-order terms and one high-order term of the matrix polynomial:

$$\tilde{H}_{GFMA} = w_0 I + w_1 \overline{A} + w_K \overline{A}^K \tag{4}$$

where $I$ is the identity matrix and $K$ is the order of the Graph-Filter. Since the coefficients $w_0, w_1, \ldots, w_K$ are learnable parameters, the filter can adaptively adjust its filtering characteristics according to the downstream task, thereby capturing more complex features. It is worth noting that in Eq. (4), computing $\overline{A}^K$ becomes computationally expensive for large values of $K$. Therefore, an efficient method is required to approximate the high-order term $\overline{A}^K$ in GFMA. To mitigate this challenge, the module employs a first-order Taylor approximation at the point $a = 1$. However, the derivative in the Taylor series is also hard to compute. So, it is further approximated using the finite difference method. By combining these two steps, the final simplified formula is derived:

$$\overline{A}^K \approx \overline{A} + (K-1)(\overline{A}^2 - \overline{A}) \tag{5}$$

This method only needs to compute $\overline{A}$ and $\overline{A}^2$, which avoids calculating high powers of the matrix, **significantly reducing the computational cost** .

### 4.2 CROSS-ATTENTION-BASED WEIGHT EMBEDDING MODEL

To more effectively capture the relationship between weights and nodes, an enhanced weight embedding model based on cross-attention is designed. The overall structure of this model is shown in Figure 2. We treat the weight vector as a special node embedding and utilize a cross-attention module to dynamically update the weight embedding at each layer of the encoder. This allows the weight information to fully interact with all node information, enabling the model to more accurately learn joint representations under different weight preferences to guide the subsequent construction process.

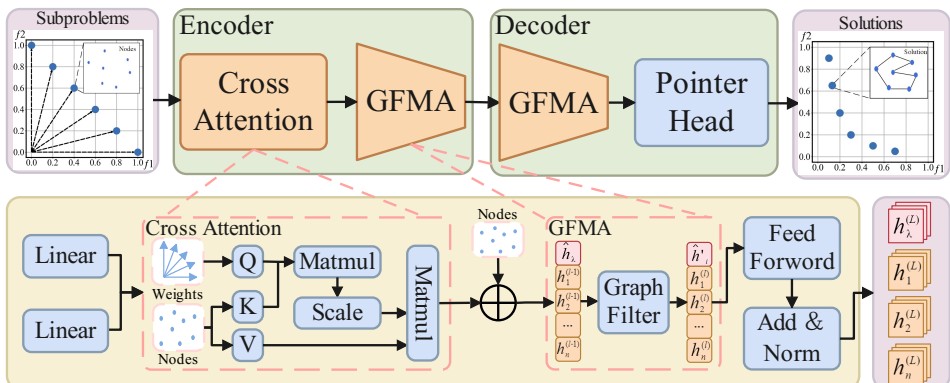

Figure 2: The overall structure of GF-MOCA.

Firstly, the weight vector $\lambda$ and the original node features $v_i$ are linearly projected into initial weight embeddings $h_\lambda^{(0)}$ and node embeddings $h_i^{(0)}$ using trainable parameter matrices $W^\lambda$, $b^\lambda$, $W^v$, and $b^v$, as shown below:

$$h_\lambda^{(0)} = W^\lambda \lambda + b^\lambda, h_i^{(0)} = W^v v_i + b^v, \forall i \in \{1, ..., n\} \tag{6}$$

In each layer $l \in \{1, ..., L\}$ of the encoder, we use a cross-attention block to update the weight embedding. The input in this module is the matrix formed by all embeddings from the previous layer $X^{(l-1)} = [h_1^{(l-1)}; ...; h_n^{(l-1)}; h_\lambda^{(l-1)}]$, where $h_\lambda^{(l-1)}$ is the last row of the matrix.

The cross-attention block normalizes the full input matrix $X^{(l-1)}$ using layer normalization, which results in $\hat{X}^{(l-1)}$. Then, $Q$, $K$, and $V$ are generated through independent linear transformations $W_q, W_k$, and $W_v$, respectively, in Eq.(7). Notably, $Q$ is generated solely from the normalized weight embedding $\hat{h}_\lambda^{(l-1)}$.

$$Q = \hat{h}_\lambda^{(l-1)} W_q, \qquad K = \hat{X}^{(l-1)} W_k, \qquad V = \hat{X}^{(l-1)} W_v \tag{7}$$

The previously generated $Q$, $K$, and $V$ are processed through an attention mechanism, which computes a weighted sum. This output is then combined with the original weight embedding $h_\lambda^{(l-1)}$ via a residual connection to produce an intermediate vector $h_\lambda'$.

$$h_\lambda' = h_\lambda^{(l-1)} + \text{softmax}\left(\frac{QK^T}{\sqrt{d_k}}\right) V \tag{8}$$

Next, the intermediate vector $h_\lambda'$ is fed into the feed-forward network (FFN) sub-layer, where it first undergoes layer normalization to produce $\hat{h}_\lambda'$. The normalized vector is passed through a Multi-Layer Perceptron (MLP) and combined with $\hat{h}_\lambda'$ via a residual connection to produce the final output

for the current layer, $h_\lambda^{(l)}$. The MLP typically consists of two linear layers and a non-linear activation function.

After obtaining the updated weight embedding $\hat{h}_\lambda$, we concatenate it with the original node embeddings $\{h_i^{(l-1)}\}$ of that layer to form an augmented set of embeddings $H'^{(l-1)} = \{h_1^{(l-1)}, \ldots, h_n^{(l-1)}, \hat{h}_\lambda\}$. The set $H'^{(l-1)}$ is fed into the GMHA sub-layer, a process that includes a residual connection and Instance Normalization, yielding intermediate embeddings $\hat{h}_i'$:

$$\hat{h}_i' = \text{IN}(h_i' + \text{GFMA}(h_i', H'^{(l-1)})), \quad \forall h_i' \in H'^{(l-1)} \tag{9}$$

Subsequently, these intermediate embeddings $\hat{h}_i'$ proceed to the FFN sub-layer. After undergoing a residual connection and instance normalization, the final representations $h_i^{(l)}$ of all embeddings output by this encoder layer are obtained as follows:

$$h_i^{(l)} = \text{IN}(\hat{h}_i' + \text{FF}(\hat{h}_i')), \quad \forall i \in \{1, ..., n\} \tag{10}$$

Through this update, we finally obtain the encoder's output weight embedding $h_\lambda^{(L)}$ and node embeddings $h_1^{(L)}, \ldots, h_n^{(L)}$. In the decoder, the node embeddings are used to calculate node selection probabilities in an auto-regressive manner over $T$ steps. In a decoding step $t \in \{1, ..., T\}$, a "glimpse" vector $q_c$ of the context embedding $h_c$ is generated by the GMHA layer introduced in Section 3.2, and then the compatibility $\alpha$ is calculated as follows:

$$\alpha_i = \begin{cases} -\infty, & \text{if node } i \text{ is masked} \\ C \cdot \tanh\left(\dfrac{q_c^T(W^K h_i^{(L)})}{\sqrt{d}}\right), & \text{otherwise} \end{cases} \tag{11}$$

where $C$ is set to 10 as Kool et al. (2018). Finally, by applying the Softmax function to the compatibility scores $\alpha$ for all nodes, we obtain the probability distribution $P_\theta(\pi_t|\pi_{1:t-1}, \lambda, s)$ to select the next node for the given weight $\lambda$ and the problem instance $s$:

$$P_\theta(\pi_t|\pi_{1:t-1}, \lambda, s) = \text{Softmax}(\alpha) \tag{12}$$

The decoder selects the next node based on this probability distribution and repeats the process until a complete solution $\pi$ is constructed for the scalarized subproblem.

## 5 EXPERIMENTS

### 5.1 EXPERIMENTAL SETTINGS

**Training Settings.** We trained our models on several different problems, including the Multi-objective Traveling Salesman Problem (MOTSP), the Multi-objective Capacitated Vehicle Routing Problem (MOCVRP), and the Multi-objective Knapsack Problem (MOKP). The problem sizes are: for MOTSP and MOCVRP, we use $n = 20/50/100$, and for MOKP, we use $n = 50/100/200$. Each model was trained for 200 epochs, with each epoch processing 100,000 random instances and a batch size $B$ of 64. The most important hyperparameter in the cross-attention model, $K$, was set to 4 (see Appendix B for more details). All experiments are implemented in Python. All experiments were conducted on a platform equipped with an NVIDIA H20 96GB GPU and an AMD EPYC 9K84 CPU.

**Baselines.** To comprehensively evaluate the performance of our method, we selected several baselines from three major categories, i.e., the single-model and multi-model neural MOCO methods, and the traditional MOEAs, for experimental comparisons. Unless otherwise specified, all compared methods use the parameter settings and model implementations reported in their original papers. Single-model MOCO methods include **PMOCO** Lin et al. (2022), and the latest SOTA methods **CNH** Fan et al. (2024) and **WE-CA** Chen et al. (2025). Multi-model MOCO methods include **DRL-MOA** Li et al. (2020), **MDRL** Zhang et al. (2022), and **EMNH** Chen et al. (2023). The **DRL-MOA** independently trains $N$ POMO models for $N$ subproblems. Particularly, DRL-MOA trains N POMO models for N subproblems, with 200 epochs for the first one and 5 epochs

for each remaining one via parameter transfer. **MDRL** Zhang et al. (2022) and **EMNH** Chen et al. (2023) both start from a unified pre-trained model and are fine-tuned for all subproblems. Traditional MOEAs include **MOEA/D** Zhang & Li (2007) and **NSGA-II** Deb et al. (2002), both implemented with 4,000 iterations, are representative decomposition-based and dominance-based MOEAs, respectively. Furthermore, we also compare with two MOCO-specific MOEAs, **MOGLS** Jaszkiewicz (2002) and **PPS/D-C** Shi et al. (2022). The **MOGLS** runs for 4000 iterations, where each iteration includes 100 local search steps, and **PPS/D-C** runs for 200 iterations. We also conduct comparisons with high-performance methods, namely, WS-LKH that uses the LKH solver on MOTSP and MOCVRP, and WS-DP that uses a dynamic programming solver on MOKP.

Table 1: Comparisons on Bi-KPs and Tri-TSPs

| Method | Bi-KP50 | | | Bi-KP100 | | | Bi-KP200 | | |
|---|---|---|---|---|---|---|---|---|---|
| | HV | Gap | Time | HV | Gap | Time | HV | Gap | Time |
| WS-DP | 0.3561 | 0.03% | 22m | 0.4532 | 0.04% | 2h | 0.3601 | 0.03% | 5.8h |
| MOEA/D | 0.3540 | 0.62% | 1.6h | 0.4508 | 0.57% | 1.7h | 0.3581 | 0.58% | 1.8h |
| NSGA-II | 0.3547 | 0.42% | 7.8h | 0.4520 | 0.31% | 8.0h | 0.3590 | 0.33% | 8.4h |
| MOGLS | 0.3540 | 0.62% | 5.8h | 0.4510 | 0.53% | 10h | 0.3582 | 0.56% | 18h |
| PPLS/D-C | 0.3528 | 0.95% | 18m | 0.4480 | 1.19% | 47m | 0.3541 | 1.69% | 1.3h |
| DRL-MOA | 0.3559 | 0.08% | 8s | 0.4531 | 0.07% | 15s | 0.3601 | 0.03% | 32s |
| MDRL | 0.3530 | 0.90% | 7s | 0.4532 | 0.04% | 18s | 0.3601 | 0.03% | 35s |
| EMNH | 0.3561 | 0.03% | 7s | **0.4535** | -0.02% | 17s | **0.3603** | -0.03% | 48s |
| PMOCO | 0.3552 | 0.28% | 8s | 0.4523 | 0.24% | 22s | 0.3595 | 0.19% | 50s |
| CNH | 0.3556 | 0.17% | 16s | 0.4527 | 0.15% | 23s | 0.3598 | 0.11% | 55s |
| WE-CA | 0.3558 | 0.11% | 8s | 0.4531 | 0.07% | 16s | 0.3602 | 0.00% | 50s |
| **GF-MOCA** | **0.3562** | 0.00% | 11s | 0.4534 | 0.00% | 26s | 0.3602 | 0.00% | 2m |

| Method | Tri-TSP20 | | | Tri-TSP50 | | | Tri-TSP100 | | |
|---|---|---|---|---|---|---|---|---|---|
| | HV | Gap | Time | HV | Gap | Time | HV | Gap | Time |
| WS-LKH | **0.4712** | 0.00% | 12m | **0.4440** | -0.18% | 1.9h | **0.5076** | -0.89% | 6.6h |
| MOEA/D | 0.4702 | 0.21% | 1.9h | 0.4314 | 2.66% | 2.2h | 0.4511 | 10.34% | 2.4h |
| NSGA-II | 0.4238 | 10.06% | 7.1h | 0.2858 | 35.51% | 7.5h | 0.2824 | 43.87% | 9.0h |
| MOGLS | 0.4701 | 0.23% | 1.5h | 0.4211 | 4.99% | 4.1h | 0.4254 | 15.44% | 13h |
| PPLS/D-C | 0.4698 | 0.30% | 1.4h | 0.4174 | 5.82% | 3.9h | 0.4376 | 13.02% | 14h |
| DRL-MOA | 0.4699 | 0.28% | 6s | 0.4303 | 2.91% | 9s | 0.4806 | 4.47% | 18s |
| MDRL | 0.4699 | 0.28% | 5s | 0.4317 | 2.59% | 10s | 0.4852 | 3.56% | 17s |
| EMNH | 0.4699 | 0.28% | 5s | 0.4324 | 2.44% | 10s | 0.4866 | 3.28% | 17s |
| PMOCO | 0.4693 | 0.40% | 5s | 0.4315 | 2.64% | 12s | 0.4858 | 3.44% | 33s |
| CNH | 0.4698 | 0.30% | 10s | 0.4358 | 1.67% | 14s | 0.4931 | 1.99% | 25s |
| WE-CA | 0.4707 | 0.11% | 5s | 0.4389 | 0.97% | 8s | 0.4975 | 1.11% | 17s |
| **GF-MOCA** | **0.4712** | 0.04% | 6s | 0.4393 | 0.88% | 13s | 0.4979 | 1.03% | 23s |
| MDRL-Aug | **0.4712** | 0.00% | 4.2m | 0.4408 | 0.54% | 25m | 0.4958 | 1.45% | 1.6h |
| EMNH-Aug | **0.4712** | 0.00% | 4.2m | 0.4418 | 0.32% | 25m | 0.4973 | 1.15% | 1.6h |
| PMOCO-Aug | **0.4712** | 0.00% | 4.9m | 0.4409 | 0.52% | 28m | 0.4956 | 1.49% | 1.6h |
| CNH-Aug | 0.4704 | 0.17% | 8.5m | 0.4409 | 0.52% | 28m | 0.4996 | 0.70% | 1.6h |
| WE-CA-Aug | **0.4712** | 0.00% | 8.2m | 0.4430 | 0.05% | 29m | 0.5029 | 0.04% | 1.7h |
| **GF-MOCA-AUG** | **0.4712** | 0.00% | 8.9m | 0.4432 | 0.00% | 33m | 0.5031 | 0.00% | 2h |

**Metrics.** Three metrics are used to evaluate all compared methods: The average Hypervolume (HV) Pan et al. (2025), gaps with respect to GF-MOCA AUG , and the total runtime for 200 instances. The Wilcoxon rank-sum test at a 1% significance level is conducted to assess statistical significances. The best results and those with no significant differences from the best are marked in **bold**, while the second-best and those statistically similar ones are underlined.

### 5.2 EXPERIMENTAL RESULTS

**Comparative Experiments.** The comparative experiments are shown in Table 1 and Table 2. We can see that GF-MOCA outperforms the WE-CA method in most experiments, especially showing a significant lead on three-objective TSP problems of size 50 and 100. However, a closer examination of the data reveals that neural combinatorial optimization methods with data augmentation incur substantial time overhead. For larger-scale problems, their computation time approaches the order of magnitude of traditional heuristic methods. Furthermore, this approach significantly increases GPU memory consumption, increasing the risk of out-of-memory (OOM) errors. Consequently, the application of data augmentation is severely limited in many resource-constrained scenarios, making solution quality without data augmentation a crucial performance indicator. Our method excels under this condition. Particularly, on the two-objective TSP problem of size 20, the GF-MOCA method without instance augmentation surpasses other baselines that use instance augmentation. This un-

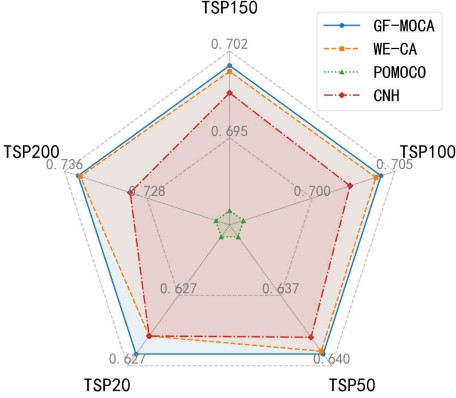

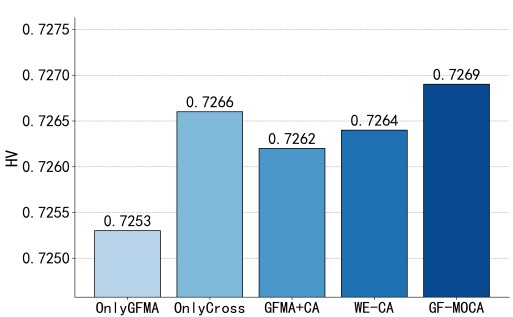

Figure 3: Comparisons Across Problem Sizes.

Figure 4: Comparisons among variants of GF-MOCA on the TSP200 instances.

derscores the significant advantage of our method's performance over its counterparts when data augmentation is not applied. Compared to traditional non-learnable heuristic methods that require longer computation times (1.6 hours), our GF-MOCA method only requires 1.4s, yet still achieves a leading position.

**Generalization Experiments on Models Trained Across Problem Sizes.** To evaluate the generalization performance of our method, we trained the mode on problem instances where the size spans the range of $\{20, 21, \ldots, 100\}$. The trained model was then tested on problem instances with sizes of 20, 50, 100, 150, and 200. The comparative results are shown in Figure 3, from which we can see that the GF-MOCA outperforms all compared methods across all tested instances. The complete test results are provided in Appendix D. In addition, the out-of-distribution generalization performance of the trained Bi-TSP model is tested on the kroab100, kroab150, and kroab200 problems Lust & Teghem (2010). The resulting Pareto fronts are shown in Figure 5. It can be observed that many solutions obtained by GF-MOCA dominate those from WE-CA and CNH. Detailed test results are given in Appendix C. These results indicate that our method is robust and generalizable, consistently performing well on different problem scales and distributions.

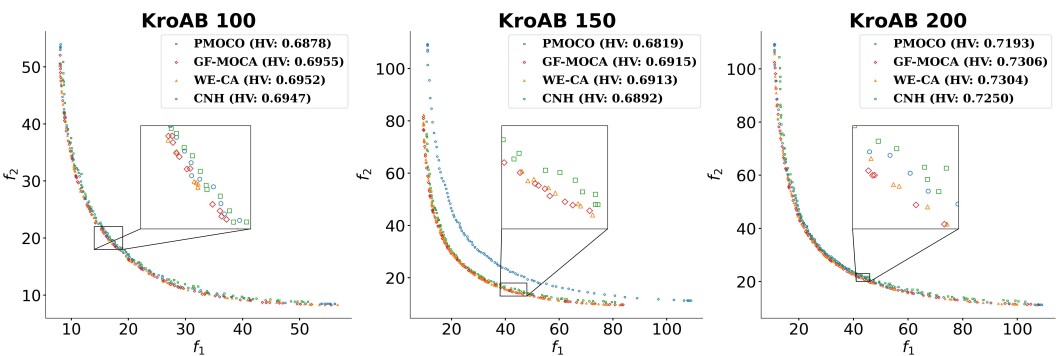

Figure 5: The obtained Pareto fronts on KroAB 100, 150, and 200 instances.

**Effectiveness of the GFMA and Cross-Attention Modules.** To evaluate the effectiveness of the GFMA and cross-attention modules, we trained several variants of the GF-MOCA, i.e., a model with only the GFMA module (OnlyGFMA), a model with only the cross-attention module (OnlyCross), and a model that uses the GFMA module to replaces the MHA module in WE-CA Chen et al. (2025) with GFMA module (GFMA+CA). All these models are tested on the TSP instances of size 200, and the comparisons with the WE-CA and GF-MOCA are shown in Figure 4. We can see that the

OnlyCross variant outperforms all except GF-MOCA and superior to WE-CA. This demonstrates the module's ability to mitigate the problems of noise and repetitive information flow inherent in WE-CA's weight embedding approach. In contrast, the performance of the OnlyGFMA variant is barely satisfactory, indicating that using only the GFMA module does not have a remarkable effect when the model is not overly complex. This can also be verified by comparing the results of GFMA-CA and WE-CA. However, the GF-MOCA eventually achieves the best results, illustrating that it is the synergy of cross-attention and GFMA that contributes to GF-MOCA's good performance.

**Hyperparameter Study.** The previous work Choi et al. (2024) shows that different $K$ values can significantly affect the performance of the GFMA module. Therefore, the affect of $K$ in GFMA is verified on size 20 BiTSP, BiCVRP, Tri-TSP, and Bi-KP. The detailed results can be found in Appendix B. According to these results, the best performance is often achieved when $K = 4$. Although for some problems, higher values such as $K = 6$ achieve the same performance as $K = 4$, we ultimately set $K = 4$ for efficiency considerations.

Table 2: Comparisons on Bi-TSPs and Bi-CVRPs

| Method | Bi-TSP20 | | | Bi-TSP50 | | | Bi-TSP100 | | |
|---|---|---|---|---|---|---|---|---|---|
| | HV | Gap | Time | HV | Gap | Time | HV | Gap | Time |
| WS-LKH | 0.6270 | 0.03% | 10m | **0.6415** | -0.02% | 1.8h | **0.7090** | -0.35% | 6h |
| MOEA/D | 0.6241 | 0.49% | 1.7h | 0.6316 | 1.53% | 1.8h | 0.6899 | 2.35% | 2.2h |
| NSGA-II | 0.6258 | 0.22% | 6.0h | 0.6120 | 4.58% | 6.1h | 0.6692 | 5.28% | 6.9h |
| MOGLS | **0.6279** | -0.11% | 1.6h | 0.6330 | 1.31% | 1.7h | 0.6854 | 2.99% | 11h |
| PPLS/D-C | 0.6256 | 0.26% | 26m | 0.6282 | 2.06% | 2.8h | 0.6844 | 3.13% | 11h |
| DRL-MOA | 0.6257 | 0.24% | 5s | 0.6360 | 0.84% | 9s | 0.6970 | 1.34% | 16s |
| MDRL | 0.6271 | 0.02% | 6s | 0.6364 | 0.78% | 8s | 0.6969 | 1.36% | 14s |
| EMNH | 0.6271 | 0.02% | 5s | 0.6364 | 0.78% | 8s | 0.6969 | 1.36% | 15s |
| PMOCO | 0.6259 | 0.21% | 6s | 0.6351 | 0.98% | 12s | 0.6957 | 1.53% | 26s |
| CNH | 0.6270 | 0.03% | 13s | 0.6387 | 0.42% | 16s | 0.7019 | 0.65% | 33s |
| WE-CA | 0.6270 | 0.03% | 6s | 0.6392 | 0.34% | 9s | 0.7034 | 0.44% | 18s |
| **GF-MOCA** | 0.6272 | -0.08% | 7s | 0.6393 | 0.33% | 14s | 0.7037 | 0.40% | 27s |
| MDRL-Aug | 0.6271 | 0.02% | 47s | 0.6408 | 0.09% | 1.8m | 0.7022 | 0.61% | 5.4m |
| EMNH-Aug | 0.6271 | 0.02% | 46s | 0.6408 | 0.09% | 1.8m | 0.7023 | 0.59% | 5.4m |
| PMOCO-Aug | 0.6270 | 0.03% | 39s | 0.6395 | 0.30% | 1.7m | 0.7016 | 0.69% | 5.8m |
| CNH-Aug | 0.6271 | 0.02% | 1.3m | 0.6410 | 0.06% | 3.9m | 0.7054 | 0.16% | 12m |
| WE-CA-Aug | 0.6271 | 0.02% | 1.3m | 0.6413 | 0.02% | 3.6m | 0.7066 | -0.01% | 12m |
| **GF-MOCA-AUG** | 0.6272 | 0.00% | 2.2m | 0.6414 | 0.00% | 5.0m | 0.7065 | 0.00% | 14m |

| Method | MOCVRP20 | | | MOCVRP50 | | | MOCVRP100 | | |
|---|---|---|---|---|---|---|---|---|---|
| | HV | Gap | Time | HV | Gap | Time | HV | Gap | Time |
| MOEA/D | 0.4255 | 1.07% | 2.3h | 0.4000 | 2.53% | 2.9h | 0.3953 | 3.16% | 5.0h |
| NSGA-II | 0.4275 | 0.60% | 6.4h | 0.3896 | 5.07% | 8.8h | 0.3620 | 11.32% | 9.4h |
| MOGLS | 0.4278 | 0.53% | 9.0h | 0.3894 | 2.92% | 20h | 0.3875 | 5.07% | 72h |
| PPLS/D-C | 0.4287 | 0.33% | 1.6h | 0.4007 | 2.36% | 9.7h | 0.3946 | 3.33% | 38h |
| DRL-MOA | 0.4287 | 0.33% | 8s | 0.4076 | 0.68% | 12s | 0.4055 | 0.66% | 21s |
| MDRL | 0.4291 | 0.23% | 6s | 0.4083 | 0.54% | 13s | 0.4056 | 0.64% | 22s |
| EMNH | 0.4299 | 0.05% | 7s | 0.4098 | 0.15% | 12s | 0.4072 | 0.24% | 22s |
| PMOCO | 0.4267 | 0.79% | 6s | 0.4036 | 1.66% | 12s | 0.3913 | 4.14% | 22s |
| CNH | 0.4287 | 0.33% | 11s | 0.4087 | 0.41% | 15s | 0.4065 | 0.42% | 25s |
| WE-CA | 0.4290 | 0.26% | 7s | 0.4089 | 0.37% | 10s | 0.4068 | 0.34% | 21s |
| **GF-MOCA** | 0.4295 | 0.16% | 8s | 0.4092 | 0.29% | 17s | 0.4073 | 0.22% | 46s |
| MDRL-Aug | 0.4294 | 0.16% | 12s | 0.4092 | 0.29% | 36s | 0.4072 | 0.24% | 2.8m |
| EMNH-Aug | **0.4302** | -0.02% | 12s | **0.4106** | -0.05% | 35s | 0.4079 | 0.07% | 2.8m |
| PMOCO-Aug | 0.4294 | 0.16% | 14s | 0.4080 | 0.58% | 42s | 0.3969 | 2.77% | 2.0m |
| CNH-Aug | 0.4299 | 0.05% | 21s | 0.4101 | 0.07% | 45s | 0.4077 | 0.12% | 1.9m |
| WE-CA-Aug | 0.4300 | 0.02% | 15s | 0.4103 | 0.02% | 36s | 0.4081 | 0.02% | 1.8m |
| **GF-MOCA-AUG** | 0.4301 | 0.00% | 24s | 0.4104 | 0.00% | 1.2m | **0.4082** | 0.00% | 4m |

## 6 CONCLUSION

This paper proposes a novel neural MOCO framework, GF-MOCA. Its cross-attention weight embedding model can more effectively capture the relationship between weights and nodes. Additionally, by incorporating the GFMA module, the over-smoothing issue induced by complex encoders can be mitigated, thereby preserving key graph structures. Comprehensive evaluations on benchmarks like MOTSP, MOCVRP, and MOKP demonstrate that GF-MOCA surpasses the performance of state-of-the-art neural MOCO methods. Future research could explore integrating advanced constraint-handling techniques, which may further enhance the scalability of our approach to real-world large-scale MOCO applications.

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

# A  MULTI-OBJECTIVE COMBINATORIAL OPTIMIZATION PROBLEMS

This appendix elaborates on the mathematical definitions of three typical Multi-Objective Combinatorial Optimization Problems involved in this study: the Multi-Objective Traveling Salesman Problem (MOTSP), the Multi-Objective Vehicle Routing Problem (MOCVRP), and the Multi-Objective Knapsack Problem (MOKP).

## A.1  MULTI-OBJECTIVE TRAVELING SALESMAN PROBLEM (MOTSP)

An instance $G$ of a MOTSP with $n + 1$ nodes is defined as follows. Each node $j \in \{0, \ldots, n\}$ has a $2\kappa$-dimensional feature vector $\boldsymbol{o}_j = [\mathrm{loc}_j^1, \mathrm{loc}_j^2, \ldots, \mathrm{loc}_j^\kappa]$, where $\mathrm{loc}_j^i \in \mathbb{R}^2$ are the 2D coordinates of node $j$ under the $i$-th objective.

The objective of the problem is to find a tour that visits all nodes $\pi = (\pi_0, \pi_1, \ldots, \pi_n)$ and simultaneously minimizes all objectives $\kappa$.his set of objective functions can be expressed as follows:

$$\min_{\pi \in X} \boldsymbol{F}(\pi) = \min(f_1(\pi), f_2(\pi), \ldots, f_\kappa(\pi)) \tag{13}$$

where the $i$-th objective function $f_i(\pi)$ is defined as the total Euclidean distance of the path in that objective's dimension:

$$f_i(\pi) = \|\mathrm{loc}_{\pi_n}^i - \mathrm{loc}_{\pi_0}^i\|_2 + \sum_{j=0}^{n-1} \|\mathrm{loc}_{\pi_j}^i - \mathrm{loc}_{\pi_{j+1}}^i\|_2 \tag{14}$$

The constraint is that the solution space $X$ contains all valid tours, meaning each node must be visited exactly once.

## A.2  MULTI-OBJECTIVE VEHICLE ROUTING PROBLEM (MOCVRP)

A Multi-Objective Vehicle Routing Problem (MOCVRP) instance consists of one depot and $n$ customer nodes. Each node $j \in \{0, \ldots, n\}$ has a feature vector $\boldsymbol{o}_j = [\mathrm{loc}_j, \delta_j]$, representing its 2D coordinates and demand, respectively. Specifically, the demand of the depot node is $\delta_0 = 0$.

The problem uses a fleet of homogeneous vehicles (with capacity $Q$) starting from the depot to serve all customers. The MOCVRP studied in this paper is a bi-objective problem, aiming to simultaneously optimize the following two conflicting objectives:

- **Minimizing total travel distance**: The sum of the lengths of all vehicle routes.
- **Minimizing the makespan**: The length of the longest route among all routes.

The problem must satisfy the following constraints:

- **Customer Uniqueness**: Each customer node must be visited by exactly one vehicle.
- **Capacity Constraint**: For any vehicle route, the sum of the demands of the customers served must not exceed the vehicle capacity $Q$.
- **Closed Routes**: All vehicle routes must start from the depot and return to the depot after serving the customers.

## A.3  MULTI-OBJECTIVE KNAPSACK PROBLEM (MOKP)

A Multi-Objective Knapsack Problem (MOKP) instance consists of $n + 1$ items to choose from and a knapsack with capacity $C$. Each item $j \in \{0, \ldots, n\}$ is defined by a feature vector $\boldsymbol{o}_j = [w_j, \boldsymbol{p}_j]$, where $w_j$ is the weight of the item, and $\boldsymbol{p}_j \in \mathbb{R}^\kappa$ is a $\kappa$-dimensional profit vector, corresponding to $\kappa$ different value objectives. The obtained solution must satisfy the core capacity constraint: the total weight of all selected items cannot exceed the knapsack's capacity $C$. The specific constraints are shown as follows.

$$\sum_{j=0}^{n} w_j \cdot x_j \leq C \tag{15}$$

The MOKP studied in this paper is a bi-objective problem ($\kappa = 2$), where the goal is to select a subset of items to simultaneously maximize the total sum of two objective profits, subject to the knapsack's capacity constraint. Let the binary decision variable $x_j \in \{0, 1\}$ indicate whether item $j$ is selected. The optimization objective can be expressed as:

$$\text{maximize} \quad \boldsymbol{F}(x) = (f_1(x), f_2(x)) \tag{16}$$

where,

$$f_1(x) = \sum_{j=0}^{n} (p_j)_1 \cdot x_j \quad \text{(Maximize the sum of the first type of profit)}$$

$$f_2(x) = \sum_{j=0}^{n} (p_j)_2 \cdot x_j \quad \text{(Maximize the sum of the second type of profit)}$$

## B  HYPERPARAMETER STUDY

**Effects of K.** Figure 6 illustrates the impact of the hyperparameter $K$ within the GFSA module on performance across the four benchmark tasks. Consistent with the findings of Choi et al. Choi et al. (2024), the optimal value for $K$ generally lies between 2 and 10, with the specific optimum varying by task. Our experiments show that setting $K = 4$ strikes a robust balance, yielding strong and consistent performance across all four problems: Bi-TSP100, MOCVRP100, Bi-KP100, and Tri-TSP100.

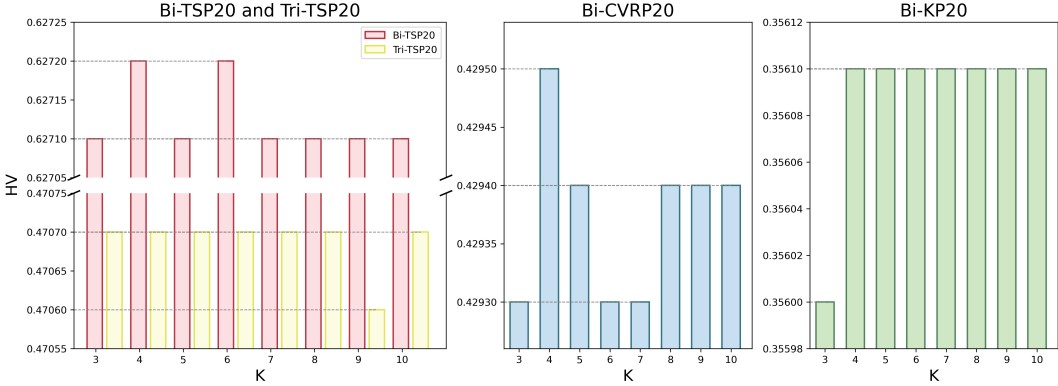

Figure 6: Effects of the $K$.

## C  DETAILED RESULTS ON OUT-OF-DISTRIBUTION INSTANCES

The detailed results of the out-of-distribution generalization are presented in Table 3. These results were obtained by testing on the bi-objective problems KroAB100, KroAB150, and KroAB200, which are synthesized from TSPLIB. Therefore, the distribution of these problems differs from the randomly generated data used during training. The results in Table 3 shows that our method significantly outperforms other neural MOCO methods in most cases, and notably surpasses all other methods when instance augmentation is not used. This demonstrates that our method also performs well on out-of-distribution instances.

## D  DETAILED RESULTS ON THE LARGE-SCALE INSTANCES

The results on the Bi-TSP150/200 instances are shown in Table 4. These results demonstrate that the GF-MOCA method also achieves excellent performance on larger-scale problems. It outperforms other neural MOCO methods on both the Bi-TSP150 and Bi-TSP200 instances, further indicating its

Table 3: Comparisons on KroAB Instances

| Method | KroAB100 | | | KroAB150 | | | KroAB200 | | |
|---|---|---|---|---|---|---|---|---|---|
| | HV | Gap | Time | HV | Gap | Time | HV | Gap | Time |
| WS-LKH | **0.7022** | -0.41% | 2.3m | **0.7017** | -0.82% | 4.0m | **0.7430** | -1.24% | 5.6m |
| MOEA/D | 0.6836 | 2.25% | 5.8m | 0.6710 | 3.59% | 7.1m | 0.7106 | 3.17% | 7.3m |
| NSGA-II | 0.6676 | 4.53% | 7.0m | 0.6552 | 5.86% | 7.9m | 0.7011 | 4.47% | 8.4m |
| MOGLS | 0.6817 | 2.52% | 52m | 0.6671 | 4.15% | 1.3h | 0.7083 | 3.49% | 1.6h |
| PPLS/D-C | 0.6785 | 2.97% | 38m | 0.6659 | 4.32% | 1.4h | 0.7100 | 3.26% | 3.8h |
| DRL-MOA | 0.6903 | 1.29% | 10s | 0.6794 | 2.39% | 12s | 0.7185 | 2.10% | 18s |
| MDRL | 0.6881 | 1.60% | 9s | 0.6831 | 1.85% | 11s | 0.7209 | 1.77% | 16s |
| EMNH | 0.6900 | 1.33% | 9s | 0.6832 | 1.84% | 11s | 0.7217 | 1.66% | 16s |
| PMOCO | 0.6878 | 1.64% | 9s | 0.6819 | 2.03% | 12s | 0.7193 | 1.99% | 17s |
| CNH | 0.6947 | 0.66% | 16s | 0.6892 | 0.98% | 19s | 0.7250 | 1.21% | 22s |
| WE-CA | 0.6952 | 0.59% | 12s | 0.6913 | 0.68% | 16s | 0.7304 | 0.48% | 23s |
| **GF-MOCA** | 0.6955 | 0.54% | 17s | 0.6915 | 0.65% | 22s | 0.7306 | 0.45% | 31s |
| MDRL-Aug | 0.6950 | 0.61% | 10s | 0.6890 | 1.01% | 16s | 0.7261 | 1.06% | 25s |
| EMNH-Aug | 0.6958 | 0.50% | 10s | 0.6892 | 0.98% | 16s | 0.7270 | 0.94% | 25s |
| PMOCO-Aug | 0.6937 | 0.80% | 11s | 0.6886 | 1.06% | 18s | 0.7251 | 1.20% | 30s |
| CNH-Aug | 0.6980 | 0.19% | 17s | 0.6938 | 0.32% | 26s | 0.7303 | 0.49% | 37s |
| WE-CA-Aug | 0.6988 | 0.07% | 19s | 0.6955 | 0.07% | 54s | 0.7343 | -0.05% | 2m |
| **GF-MOCA-AUG** | 0.6993 | 0.00% | 43s | 0.6960 | 0.00% | 2m | 0.7339 | 0.00% | 5m |

potential for solving even larger-scale challenges. Although the solution quality of GF-MOCA does not surpass the traditional WS-LKH method, it holds a significant advantage in terms of computational time.

Table 4: Comparisons on Bi-TSP150 and Bi-TSP200 Instances

| **Method** | **Bi-TSP150** | | | **Bi-TSP200** | | |
|---|---|---|---|---|---|---|
| | **HV** | **Gap** | **Time** | **HV** | **Gap** | **Time** |
| WS-LKH | **0.7149** | $-2.01\%$ | 13h | **0.7490** | $-2.00\%$ | 22h |
| MOEA/D | 0.6809 | 2.84% | 2.4h | 0.7139 | 2.78% | 2.7h |
| NSGA-II | 0.6659 | 4.98% | 6.8h | 0.7045 | 4.06% | 6.9h |
| MOGLS | 0.6768 | 3.42% | 22h | 0.7114 | 3.12% | 38h |
| PPLS/D-C | 0.6784 | 3.20% | 21h | 0.7106 | 3.23% | 32h |
| DRL-MOA | 0.6901 | 1.53% | 36s | 0.7219 | 1.69% | 1.2m |
| MDRL | 0.6922 | 1.23% | 36s | 0.7251 | 1.25% | 1.1m |
| EMNH | 0.6930 | 1.11% | 37s | 0.7260 | 1.13% | 1.1m |
| PMOCO | 0.6910 | 1.40% | 42s | 0.7231 | 1.53% | 1.3m |
| CNH | 0.6985 | 0.33% | 50s | 0.7292 | 0.69% | 1.4m |
| WE-CA | 0.7003 | 0.07% | 45s | 0.7341 | 0.03% | 1.3m |
| **GF-MOCA** | 0.7008 | 0.00% | 1.4m | 0.7343 | 0.00% | 2.4m |

# E    GENERALIZATION OF UNIFIED TRAINING ACROSS PROBLEM SIZES.

For the WE-CA, CNH, and PMOCO methods, we trained a single unified model on a range of problem sizes, specifically $n \in \{20, 21, \ldots, 100\}$ (with the exception of Bi-KP, for which the range was $n \in \{50, 51, \ldots, 200\}$). For MORAM, we used the provided pre-trained model. The suffix "-U" distinguishes the unified model trained on varying sizes, while "-n" denotes a model trained on a fixed size (e.g., "-50"). In the cases of WE-CA and PMOCO, we report only the results from

Table 5: Bi-objective and Tri-objective Results with Unified Width

| | Bi-TSP20 | | | Bi-TSP50 | | | Bi-TSP100 | | |
|---|---|---|---|---|---|---|---|---|---|
| Method | HV↑ | Gap↓ | Time↓ | HV↑ | Gap↓ | Time↓ | HV↑ | Gap↓ | Time↓ |
| MORAM | 0.6216 | 0.88% | 1s | 0.6255 | 2.48% | 2s | 0.6821 | 3.47% | 3s |
| PMOCO | 0.6270 | 0.02% | 14s | 0.6387 | 0.42% | 17s | 0.7019 | 0.67% | 29s |
| PMOCO-50 | 0.6262 | 0.14% | 6s | 0.6351 | 0.98% | 10s | 0.6915 | 2.15% | 19s |
| PMOCO-U | 0.6111 | 2.55% | 7s | 0.5939 | 7.41% | 11s | 0.6417 | 9.18% | 21s |
| WE-CA-50 | 0.6267 | 0.06% | 6s | 0.6391 | 0.36% | 10s | 0.6988 | 1.10% | 19s |
| WE-CA-U | 0.6270 | 0.02% | 7s | 0.6392 | 0.34% | 10s | 0.7034 | 0.45% | 21s |
| GF-MOCA-50 | 0.6262 | 0.14% | 6s | 0.6393 | 0.33% | 10s | 0.6993 | 1.03% | 19s |
| GF-MOCA-U | 0.6270 | 0.02% | 7s | 0.6393 | 0.33% | 10s | 0.7036 | 0.47% | 21s |
| CNH-Aug | 0.6271 | 0.00% | 1.5m | 0.6410 | 0.06% | 4.1m | 0.7054 | 0.17% | 16m |
| PMOCO-50-Aug | 0.6270 | 0.02% | 1.0m | 0.6395 | 0.30% | 3.2m | 0.6977 | 1.26% | 15m |
| PMOCO-U-Aug | 0.6253 | 0.29% | 1.0m | 0.6126 | 4.49% | 3.3m | 0.6558 | 7.19% | 15m |
| WE-CA-50-Aug | 0.6272 | -0.02% | 1.0m | 0.6412 | 0.03% | 3.3m | 0.7034 | 0.45% | 15m |
| WE-CA-U-Aug | 0.6271 | 0.00% | 1.0m | 0.6413 | 0.02% | 3.3m | **0.7066** | **0.00%** | 16m |
| GF-MOCA-50-Aug | **0.6273** | **-0.03%** | 1.0m | **0.6414** | **0.00%** | 5m | 0.7039 | 0.38% | 30m |
| GF-MOCA-U-Aug | 0.6271 | 0.00% | 1.0m | **0.6414** | **0.00%** | 5m | **0.7066** | **0.00%** | 30m |

| | Bi-CVRP20 | | | Bi-CVRP50 | | | Bi-CVRP100 | | |
|---|---|---|---|---|---|---|---|---|---|
| Method | HV↑ | Gap↓ | Time↓ | HV↑ | Gap↓ | Time↓ | HV↑ | Gap↓ | Time↓ |
| CNH | 0.4287 | 0.30% | 15s | 0.4087 | 0.39% | 17s | 0.4065 | 0.42% | 31s |
| PMOCO-50 | 0.4191 | 2.53% | 9s | 0.4036 | 1.63% | 12s | 0.4014 | 1.67% | 26s |
| PMOCO-U | 0.4275 | 0.58% | 8s | 0.4068 | 0.85% | 13s | 0.4044 | 0.93% | 25s |
| WE-CA-50 | 0.4238 | 1.44% | 7s | 0.4090 | 0.32% | 12s | 0.4025 | 1.40% | 25s |
| WE-CA-U | 0.4290 | 0.23% | 7s | 0.4089 | 0.34% | 12s | 0.4068 | 0.34% | 26s |
| GF-MOCA-50 | 0.4213 | 2.02% | 7s | 0.4092 | 0.27% | 12s | 0.4025 | 1.40% | 25s |
| GF-MOCA-U | 0.4287 | 0.30% | 7s | 0.4090 | 0.32% | 12s | 0.4068 | 0.34% | 26s |
| CNH-Aug | 0.4299 | 0.02% | 22s | 0.4101 | 0.05% | 45s | 0.4077 | 0.12% | 2.5m |
| PMOCO-50-Aug | 0.4270 | 0.70% | 15s | 0.4080 | 0.56% | 36s | 0.4051 | 0.76% | 2.4m |
| PMOCO-U-Aug | 0.4296 | 0.09% | 15s | 0.4095 | 0.19% | 40s | 0.4071 | 0.27% | 2.4m |
| WE-CA-50-Aug | 0.4277 | 0.53% | 15s | 0.4103 | 0.00% | 37s | 0.4056 | 0.64% | 2.5m |
| WE-CA-U-Aug | **0.4300** | **0.00%** | 14s | 0.4103 | 0.00% | 40s | 0.4081 | 0.02% | 2.5m |
| GF-MOCA-50-Aug | 0.4268 | 0.74% | 15s | **0.4104** | **-0.02%** | 40s | 0.4068 | 0.34% | 5m |
| GF-MOCA-U-Aug | **0.4300** | **0.00%** | 14s | 0.4103 | 0.00% | 40s | **0.4082** | **0.00%** | 4m |

| | Bi-KP50 | | | Bi-KP100 | | | Bi-KP200 | | |
|---|---|---|---|---|---|---|---|---|---|
| Method | HV↑ | Gap↓ | Time↓ | HV↑ | Gap↓ | Time↓ | HV↑ | Gap↓ | Time↓ |
| CNH | 0.3556 | 0.08% | 18s | 0.4527 | 0.11% | 27s | 0.3598 | 0.11% | 1.2m |
| PMOCO-100 | 0.3548 | 0.31% | 11s | 0.4523 | 0.20% | 19s | 0.3527 | 2.08% | 1.0m |
| PMOCO-U | 0.3503 | 1.57% | 10s | 0.4484 | 1.06% | 19s | 0.3536 | 1.83% | 1.0m |
| WE-CA-100 | **0.3559** | **0.00%** | 10s | 0.4533 | -0.02% | 19s | 0.3591 | 0.31% | 1.0m |
| WE-CA-U | 0.3558 | 0.03% | 9s | 0.4531 | 0.02% | 21s | **0.3602** | **0.00%** | 1.1m |
| GF-MOCA-100 | 0.3558 | 0.03% | 9s | **0.4534** | **-0.04%** | 22s | 0.3544 | 1.61% | 2m |
| GF-MOCA-U | **0.3559** | **0.00%** | 9s | 0.4532 | 0.00% | 25s | **0.3602** | **0.00%** | 2m |

| | Tri-TSP20 | | | Tri-TSP50 | | | Tri-TSP100 | | |
|---|---|---|---|---|---|---|---|---|---|
| Method | HV↑ | Gap↓ | Time↓ | HV↑ | Gap↓ | Time↓ | HV↑ | Gap↓ | Time↓ |
| MORAM | 0.4573 | 2.95% | 1s | 0.4101 | 7.47% | 2s | 0.4588 | 8.79% | 3s |
| CNH | 0.4698 | 0.30% | 10s | 0.4358 | 1.67% | 14s | 0.4931 | 1.97% | 26s |
| PMOCO-50 | 0.4682 | 0.64% | 6s | 0.4315 | 2.64% | 8s | 0.4824 | 4.10% | 21s |
| PMOCO-U | 0.4691 | 0.45% | 7s | 0.4318 | 2.57% | 10s | 0.4873 | 3.12% | 21s |
| WE-CA-50 | 0.4691 | 0.45% | 5s | 0.4389 | 0.97% | 9s | 0.4877 | 3.04% | 20s |
| WE-CA-U | 0.4707 | 0.11% | 5s | 0.4389 | 0.97% | 9s | 0.4975 | 1.09% | 20s |
| GF-MOCA-50 | 0.4678 | 0.72% | 5s | 0.4387 | 1.02% | 9s | 0.4906 | 2.47% | 20s |
| GF-MOCA-U | 0.4708 | 0.08% | 5s | 0.4393 | 0.88% | 9s | 0.4977 | 1.05% | 20s |
| CNH-Aug | 0.4704 | 0.17% | 8.0m | 0.4409 | 0.52% | 33m | 0.4996 | 0.68% | 2.1h |
| PMOCO-50-Aug | **0.4713** | **-0.02%** | 5.2m | 0.4409 | 0.52% | 28m | 0.4933 | 1.93% | 1.7h |
| PMOCO-U-Aug | 0.4712 | 0.00% | 5.2m | 0.4406 | 0.59% | 31m | 0.4968 | 1.23% | 1.7h |
| WE-CA-50-Aug | 0.4711 | 0.02% | 5.3m | 0.4431 | 0.02% | 29m | 0.4991 | 0.78% | 1.9h |
| WE-CA-U-Aug | 0.4712 | 0.00% | 5.2m | 0.4431 | 0.02% | 31m | 0.5027 | 0.06% | 1.8h |
| GF-MOCA-50-Aug | 0.4712 | 0.00% | 5.3m | **0.4432** | **0.00%** | 40m | 0.4994 | 0.72% | 2.5h |
| GF-MOCA-U-Aug | 0.4712 | 0.00% | 5.2m | **0.4432** | **0.00%** | 40m | **0.5030** | **0.00%** | 2.5h |

the "-50" models, as they performed significantly better than their "-20" and "-100" counterparts. The "Gap" column in the results indicates the performance difference of each method relative to GF-MOCA-U-Aug. As shown in the Table 5, among all the neural approaches, including the CNH method, our GF-MOCA-U and GF-MOCA-50 outperform other methods on nearly all problem instances. With data augmentation, our GF-MOCA-U-Aug and GF-MOCA-50-Aug variants also demonstrate superior performance in almost all cases. Notably, on the Bi-TSP20 problem, our GF-MOCA-50 method even surpasses models that were trained specifically on size-20 instances. These results indicate that our method is both robust and generalizable across diverse problem scales and distributions.

