# OpenReview forum: "Decomposition-based Neural Multi-objective Combinatorial Optimization with Graph-Filter based Multi-Head Attention"
_ICLR.cc/2026/Conference — ICLR 2026 Conference Withdrawn Submission_

### Official Review · Reviewer_kpBL · 2025-10-27

**Soundness:** 2
**Presentation:** 3
**Contribution:** 2
**Rating:** 2
**Confidence:** 4

**Summary:**

This paper presents an enhanced Transformer-based architecture for addressing decomposition-based multi-objective combinatorial optimization problems. To mitigate the noise and training instability caused by directly combining weight vectors with problem instances in prior architectures, the authors introduce the Graph-Filtered Multi-objective Cross-Attention Framework (GF-MOCA). In this framework, a Cross-Attention mechanism is employed in the encoder to update weight vectors and node features separately. Additionally, to alleviate over-smoothing issues arising from deep encoders, the authors propose a Graph-Filter-based Multi-Head Attention (GFMA) module. This module maintains computational efficiency while effectively addressing over-smoothing. Experimental results demonstrate that GF-MOCA outperforms the WE-CA method in terms of solution quality and generalization capabilities.

**Strengths:**

1. The paper presents its proposed method in a clear and well-structured manner. Through detailed textual explanations and intuitive illustrations, the authors thoroughly describe the two core components—Cross-Attention (CA) and Graph-Filter-based Multi-Head Attention (GFMA). They clarify the design rationale, internal structure, and roles of these modules within the framework. As a result, readers can easily grasp the core ideas and architectural logic of the proposed method.

2. Compared with the state-of-the-art WE-CA method, GF-MOCA exhibits competitive or superior performance across diverse problem settings. The experimental design is extensive, covering multiple representative tasks and datasets. Furthermore, ablation studies validate the contributions and effectiveness of the CA and GFMA modules.

**Weaknesses:**

1. While the proposed Cross-Attention module builds on existing concepts, it does not introduce significant innovation. Although the introduction of the Graph Filter may be novel in this context, similar ideas have appeared in prior works, such as "Graph Convolutions Enrich the Self-Attention in Transformers." While the authors cleverly reduce the computational complexity of GFMA using a Taylor expansion, the overall architectural innovation of the proposed method remains limited.

2. The paper does not adequately explain the root causes of over-smoothing. It merely states that increased encoder complexity can lead to over-smoothing, without providing theoretical analysis or relevant references. Additionally, in the ablation study, the performance gap between the "OnlyCross" variant and the final GF-MOCA model (with the graph filter) is not particularly significant. This leaves the discussion and empirical evidence regarding this critical issue relatively unconvincing.

3. The performance gains of GF-MOCA over WE-CA appear modest. While GF-MOCA achieves slightly better results in reported tables, the differences are on the order of 0.1%, which may not be sufficient to demonstrate meaningful practical improvements. Given that GF-MOCA is directly inspired by WE-CA, a clearer justification of its actual performance gains would strengthen the contribution.

**Questions:**

1. The experiments in the paper focus on problem instances with sizes up to 200. Can the proposed method handle larger-scale problems? If so, how does its performance scale with increasing problem size?

2. The over-smoothing issue is frequently mentioned in the paper. Could the authors provide additional experiments or references to demonstrate that using Cross-Attention alone indeed leads to over-smoothing in practice?

3. Sections 3 and 4 lack sufficient citations. Since both the Cross-Attention and Graph Filter components are built upon existing concepts, could the authors include more relevant references to strengthen the methodological foundation of the paper?

---

### Official Review · Reviewer_94dD · 2025-10-30

**Soundness:** 1
**Presentation:** 2
**Contribution:** 1
**Rating:** 2
**Confidence:** 4

**Summary:**

This paper introduces GF-MOCA, a neural multi-objective combinatorial optimization framework designed to better model weight-instance relationships through a cross-attention-based weight embedding model and a Graph-Filter-based Multi-Head Attention (GFMA) module. Although the method shows some improvements in experimental results, there are several issues with the motivation, innovation, and analysis that need to be addressed for a more thorough evaluation.

**Strengths:**

1. The paper proposes a cross-attention mechanism for modeling weight-node interactions in neural multi-objective combinatorial optimization.
2. The method is evaluated on a range of multi-objective combinatorial optimization problems, and the results demonstrate some improvements over existing methods.

**Weaknesses:**

1. The paper mentions solving the over-smoothing problem existed in previous neural multi-objective combinatorial optimization approaches but does not adequately explain what over-smoothing is, nor does it validate how the proposed method effectively addresses this issue. A more detailed discussion and verification are required.
2. The core contribution of the paper appears to be a modification of the attention mechanism, which feels more like an incremental improvement rather than a significant innovation. The novelty should be better highlighted and justified.
3. Some recent studies on neural multi-objective combinatorial optimization, such as [1], are not discussed or compared in the paper. A more comprehensive literature review would strengthen the argument and context for this research.
4. While the method shows some improvements over WE-CA, the performance gains are modest. The paper could further justify the significance of these small improvements.
5. There is no comparison of computational costs, such as parameter count and inference time, between the proposed method and previous methods. A discussion of these aspects would provide a clearer picture of the method's efficiency.
6. The paper lacks adequate ablation experiments to validate the effectiveness of individual components of GF-MOCA.
7. There are several writing mistakes throughout the paper, such as inconsistent hyphenation, inconsistent capitalization, and inconsistent term (e.g., PPS/D-C vs PPLS/D-C).
8. The citation formatting in the paper is incorrect. For example, “Ahmadi et al. (2016)” should be “(Ahmadi et al. 2016)”.
9. In the Metrics section, the paper cites a seemingly unrelated reference [Pan et al., 2025] when discussing the hypervolume metric.

[1] Neural Multi-Objective Combinatorial Optimization via Graph-Image Multimodal Fusion. ICLR 2025.

**Questions:**

See Weaknesses.

---

### Official Review · Reviewer_BNyN · 2025-11-01

**Soundness:** 2
**Presentation:** 2
**Contribution:** 2
**Rating:** 2
**Confidence:** 4

**Summary:**

This work addresses decomposition-based neural methods for multi-objective combinatorial optimization. It observes that recent weight-specific models (e.g., CNH, WE-CA) still struggle to reliably fuse node features with the weight vector; directly mixing the two (as in conditional attention) can blur node information and introduce instability. The proposed GF-MOCA separates weights and nodes into distinct sequences, updating weight embeddings via cross-attention, and augments the encoder with a Graph-Filter Multi-Head Attention (GFMA) that blends identity, local, and higher-order interactions to curb over-smoothing in deeper stacks while remaining computationally efficient through a simple approximation. Experiments on MOTSP, MOCVRP, and MOKP report competitive or superior hypervolume against strong neural and classical baselines, with the clearest gains on tri-objective TSP at sizes n=50/100. The method remains strong without instance augmentation and shows out-of-distribution robustness on TSPLIB KroAB instances.

**Strengths:**

Clear motivation and design – The paper precisely identifies the fusion bottleneck and responds with a clean two-sequence + cross-attention design; queries are produced solely from weights, which is easy to reason about and reproduce.

Simple, adaptable GFMA – The polynomial graph filter GFMA with learnable coefficients is concise and hardware-friendly; the A^K approximation reduces higher-order costs while preserving structure.

Breadth of evaluation – The study covers Tri/Bi-TSP, Bi-CVRP, and Bi-KP, includes ablations (OnlyCross, OnlyGFMA, GFMA+CA), and reports OOD tests on TSPLIB KroAB.

Robust performance without augmentation – The paper emphasizes that GF-MOCA remains strong without instance augmentation, which is costly and risks OOM.

**Weaknesses:**

[Effect sizes vs. narrative]
Several improvements over WE-CA are marginal or tied—especially on Bi-TSP and Bi-CVRP—while the text occasionally generalizes to a “significant lead.” This can overstate practical impact when differences fall within noise or rounding. The Tri-TSP highlights are convincing, but the broader summary should more accurately reflect small effect sizes elsewhere.

[Evidence for WE-CA drawbacks]
The alleged issues with WE-CA—node-feature distortion, noise injection, and training instability—are not empirically validated with stability or representation-quality metrics. Although the OnlyCross variant outperforming WE-CA suggests a benefit, the underlying mechanism remains inferential rather than demonstrated.

[GFMA and over-smoothing diagnostics]
GFMA is motivated as addressing over-smoothing in deeper encoders, yet the paper provides no direct diagnostics (e.g., layer-wise embedding similarity, variance preservation, cluster separability) that link GFMA to reduced smoothing. As a result, the causal relationship is asserted but not shown.

[Scope of generalization]
The generalization narrative reads broader than the evidence presented: size generalization and TSP-family OOD robustness are strong, but cross-type generalization (e.g., training on TSP and testing on CVRP/KP) is not reported, so claims should be scoped accordingly.

**Questions:**

Suggestions for Improvement

[Report effect sizes and statistical evidence consistently]
Please attach effect sizes (absolute and relative HV differences), paired tests (e.g., Wilcoxon p-values), and dispersion (CIs or std) alongside each table; then rewrite summary phrases (“significant lead”) to mirror the actual magnitudes, especially where differences are marginal or tied.

[Substantiate the WE-CA diagnosis]
Add diagnostics that quantify “instability/noise,” such as loss/gradient variance over training, failure/rollback rates, or representation preservation scores for node features; show that the OnlyCross > WE-CA advantage co-varies with these metrics.

[Provide mechanistic evidence for over-smoothing relief]
Measure layer-wise embedding similarity (e.g., cosine or CKA), feature variance, and cluster separability with/without GFMA, ideally in a 2×2 (Cross-Attn × GFMA) design, to demonstrate that gains arise from reduced smoothing.

[Isolate GFMA efficiency]
Report GFMA’s FLOPs/wall-clock/memory against a non-approximated A^K baseline under a controlled setting (no augmentation, fixed batch/hardware); include a microbenchmark table so readers can attribute net savings to the filter.

[Clarify the scope of generalization]
Either add cross-type tests (e.g., train on TSP, zero-shot or few-shot on CVRP/KP) or explicitly limit the claim to size and TSP-family OOD generalization to align text with evidence.

[Strengthen ablation rigor and reproducibility]
For OnlyCross/OnlyGFMA/GFMA+CA, include dispersion and paired tests; also consolidate seeds, steps, batch sizes, learning rates, hardware, and wall-clock per method in a single reproducibility table.

[Summarize hyperparameter guidance]
Surface the K sensitivity (currently in the appendix) in the main text and provide a recommended range (e.g., default K=4) per task to aid practitioners.

---

### Official Review · Reviewer_jnqJ · 2025-11-01

**Soundness:** 3
**Presentation:** 3
**Contribution:** 2
**Rating:** 4
**Confidence:** 3

**Summary:**

This paper presents GF-MOCA (Graph-Filtered Multi-objective Cross-Attention), a novel learning-based framework for solving Multi-Objective Combinatorial Optimization Problems (MOCOPs) via a decomposition-based neural approach. The work aims to address two key limitations of existing state-of-the-art (SOTA) methods (e.g., CNH and WE-CA): (1) the ineffective fusion of weight vector (λ) and instance node information, and (2) the risk of over-smoothing and instability in complex deep encoders.

The proposed GF-MOCA framework introduces two core components:
- Cross-Attention-based Weight Embedding Module: This module treats the weight vector λ and the node vectors as two separate sequences, employing a tailored cross-attention mechanism where the query is generated only from the weight embedding. This design is intended to achieve a more profound, yet controlled, understanding of the weight-instance inter-dependencies.
- Graph-Filter-based Multi-Head Attention (GFMA) Module: This module integrates a graph filter composed of an identity term and matrix polynomials into the attention calculation. This is specifically designed to mitigate the over-smoothing problem inherent in deep GNN-like architectures, preserving critical structural information. The paper uses an efficient Taylor approximation to simplify the high-order matrix polynomial computation.

Extensive experiments on classical MOCOP benchmarks, including MOTSP, MOCVRP, and MOKP (Bi- and Tri-objective variants), demonstrate that GF-MOCA significantly outperforms existing neural MOCO methods in terms of Hypervolume (HV) and exhibits strong generalization capabilities across problem scales and out-of-distribution instances.

**Strengths:**

- Superior Performance and Generalization: GF-MOCA consistently outperforms all competing SOTA neural MOCO methods (CNH, WE-CA, PMOCO) across MOTSP, MOCVRP, and MOKP benchmarks, often achieving performance comparable to or better than computationally expensive traditional high-performance heuristics (WS-LKH, WS-DP), especially considering its fast inference time. Furthermore, the strong results on out-of-distribution (KroAB) and unified training across different problem sizes highlight its robustness.
- Elegant and Effective Methodological Design: The proposed two-pronged approach (Cross-Attention for enhanced weight-instance fusion, and GFMA for encoder stability/over-smoothing mitigation) is conceptually strong and empirically validated by the ablation study.
- Comprehensive Experimental Validation: The evaluation is extensive, including a wide array of problems, competitive baselines, and crucial ablation studies to justify the role of each component.

**Weaknesses:**

- Limited Component Novelty of GFMA: The GFMA module is explicitly based on the work of Choi et al. (2024), utilizing the same graph-filter approximation (Eq 5). While its integration and justification for mitigating over-smoothing in this specific MOCO context are novel, the design of the GFMA component itself is adapted. The authors should be more explicit in the introduction/methods about the level of reuse to clearly delineate their novelty.
- Computational Bottleneck of Data Augmentation: The best performing variant, GF-MOCA-AUG, incurs a substantial time overhead (e.g., Tri-TSP100: 2.5h for GF-MOCA-AUG vs. 20s for GF-MOCA). The paper highlights the issue, but a deeper analysis or a proposal for a more efficient augmentation strategy (e.g., selective sampling or a different loss function to substitute augmentation) would significantly improve the practical utility of the best model variant. The current gap limits the practicality of the SOTA performance.
- Empirical Over-smoothing Evidence: The claim of mitigating over-smoothing is plausible and supported by the ablation. However, the evidence is purely a performance score. The paper would be strengthened by a more direct, visual, or quantitative analysis of how GFMA affects the node embeddings (e.g., layer-wise distance metrics of embeddings) compared to a standard MHA to definitively demonstrate the over-smoothing mitigation mechanism in action.

**Questions:**

- **Generalization to Many-Objective Problems (More than 3 Objectives)**: The current framework uses the Weighted Sum (WS) decomposition, which is known to struggle with non-convex Pareto fronts, and the experiments are limited to Bi- and Tri-objective problems. Can the authors comment on:
    1.  The expected performance and potential limitations of **GF-MOCA** on problems with a larger number of objectives (e.g., five objectives)?
    2.  The feasibility and expected modifications needed to integrate **GF-MOCA** with other decomposition methods (e.g., Tchebycheff or PBI) that are often preferred for problems with more complex Pareto front shapes?

- **Computational Analysis of GFMA**: The use of the Taylor approximation (referring to the relevant equation in the paper) is meant to reduce the computational cost of the high-order term in the attention matrix. Can the authors provide a clearer analysis or empirical verification of the actual computational saving of **GFMA** compared to a full MHA implementation for different layer counts and sequence lengths $N$ (especially for large $N$)? This would better justify the engineering choice.

---

### Official Review · Reviewer_s8dY · 2025-11-01

**Soundness:** 2
**Presentation:** 3
**Contribution:** 2
**Rating:** 4
**Confidence:** 4

**Summary:**

This paper studies how to encode weight vectors for neural MOCOPs. It proposes to treat weights and nodes as separate sequences to model their interactions via cross-attention, and augment standard attention with a graph-filtered module. Evaluations are conducted on multiple benchmarks.

**Strengths:**

1. The problem studied is interesting.
2. The proposed method is evaluated on extensive benchmarks.

**Weaknesses:**

1. The paper lacks a discussion of computational complexity for the proposed GFMA module. In contrast to the standard Multi-Head Attention (MHA), which has a complexity of $O(N^2)$ (quadratic in the number of nodes $N$), GFMA introduces an additional dense matrix multiplication $A^2 = A \times A$, resulting in $O(N^3)$ time complexity. This cubic scaling can significantly slow down both training and inference as $N$ increases, especially in larger-scale scenarios (e.g., 1000 nodes), deeper encoders, or when integrating into the heavy-decoder paradigm [1, 2].
2. The performance improvement is relatively marginal compared to the extra parameters and computation introduced by the proposed method.
3. The ablation is limited to a single test result, please expand with additional evaluations (e.g., multiple TSP sizes evaluated using the same trained model) to better support the claims.
4. Training resource usage is unclear, please report a comparison of training time and peak GPU memory for GF-MOCA vs. a neural baseline (e.g., WE-CA) under identical settings.


References

[1] BQ-NCO: Bisimulation Quotienting for Efficient Neural Combinatorial Optimization.

[2] Neural Combinatorial Optimization with Heavy Decoder: Toward Large Scale Generalization.

**Questions:**

1. The current formulation of GFMA module seems requires $A$ to be a square matrix, which would limit applicability to cross-attention with rectangular attention scores. However, many COPs involve more than one set of objects (like FJSP), where cross-attention is commonly used. Can GFMA be generalized to handle rectangular cases？
2. In the GFMA module, it seems that after approximation, $w_0 I + w_1A + w_KA^K \approx w_0 I + [w_1 - w_K (K-2)] A + w_K (K-1) A^2 = w_0^\prime I + w_1^\prime A + w_{2}^\prime A^2$, i.e., it takes the same functional form as setting $K=2$ after some reparameterization. So I am curious whether the observed performance deviation across different $K$ values mainly comes from different initialization or from the reparameterization of the weights $w$.
3. Figure 2 suggests the decoder also uses a GFMA layer, whereas line 289 of the text states that GMHA is used, please address the discrepancy.

---

### Note · Authors · 2026-01-12

**Comment:**

We thank the reviewers and Area Chair for their constructive feedback. After careful consideration, we have decided to withdraw the paper at this stage to address the raised concerns and significantly revise the work. We intend to resubmit an improved version to a future venue.

**Withdrawal Confirmation:**

I have read and agree with the venue's withdrawal policy on behalf of myself and my co-authors.